# The Asian tiger mosquito *Aedes albopictus* (Skuse) in Kosovo: First record

**Nesade Muja-Bajraktari[1], Përparim Kadriaj[2], Ferdije Zhushi-Etemi[1], Kurtesh Sherifi[3], Bulent Alten[4], Dusan Petrić[5], Enkelejda Velo [2]\*, Francis Schaffner[6]**

**1** Department of Biology, Faculty of Mathematics and Natural Sciences, University" Hasan Prishtina", Prishtina, Republic of Kosovo, **2** Department of Epidemiology and Control of Infectious Diseases, Institute of Public Health, Tirana, Albania, **3** Department of Veterinary Medicine, Faculty of Agriculture and Veterinary, University" Hasan Prishtina", Prishtina, Republic of Kosovo, **4** VERG Laboratories, Ecology Division, Biology Department, Faculty of Science, Hacettepe University, Ankara, Turkey, **5** Laboratory for Medical and Veterinary Entomology, Faculty of Agriculture, University of Novi Sad, Novi Sad, Serbia, **6** Francis Schaffner Consultancy, Riehen, Switzerland

\* keladikolli@yahoo.com

**Data Availability Statement:** All relevant data are within the paper and its Supporting Information file.

**Funding:** Ovitraps used in this study were provided by ECDC VectorNet project to KS, NMB, and FS (framework contract OC/EFSA/AHAW/2013/02-

## Abstract

The Asian tiger mosquito, *Aedes albopictus*, is an invasive mosquito species that is considered a potential vector of about 22 arboviruses, among which dengue, chikungunya and Zika. Here we report the first record of *Aedes albopictus* in the territory of the Republic of Kosovo. The first finding, in July 2020, was driven by a photo of an adult mosquito published by a citizen in social media. The subsequent field investigation in July 2020 confirmed the presence of adult mosquitoes by human landing catch and collection of eggs in ovitraps at the village Zhur. Monitoring was performed for 7 weeks with ovitraps and BG-Sentinel adult traps at 36 sampling stations from 7 localities, in the Southern part of Kosovo, on the border with Albania. Fifty-two out of 81 ovitrap samples resulted positive for the presence of *Ae*. *albopictus*. A total of 2,711 eggs are collected in 22 out 36 stations and a total of 78 adults from 4 out 4 stations with BG Sentinel trap and 15 adults with handled electric aspirator. Our finding shows that the tiger mosquito is expanding its geographical range in the Balkans, southeastern Europe.

## Introduction

*Aedes* (*Stegomyia*) *albopictus* (Skuse, 1894) (Diptera: Culicidae), commonly called the 'Asian tiger mosquito', is widespread throughout the tropical and subtropical regions of the world. During the last 30 years, it has invaded many countries worldwide, including in the Mediterranean Basin. The species is currently considered one of the top 100 invasive species globally and the most widespread invasive mosquito species in Europe [1, 2]. It represents a severe threat to public health due to its aggressive daytime human-biting behavior and its vectorial competence for numerous arboviruses of the family Flaviviridae (e.g., dengue, West Nile, and Japanese encephalitis viruses), Bunyaviridae (e.g., Rift Valley fever, Potosi, Cache Valley, and La Crosse viruses), Togaviridae (e.g., chikungunya and Ross River viruses) [3–7]. After *Ae*.

FWC1 funded by the European Food Safety Authority (EFSA) and the European Centre for Disease prevention and Control (ECDC)).

**Competing interests:** The authors have declared that no competing interests exist.

*aegypti*, *Ae. albopictus* is the secondary vector of dengue and dengue hemorrhagic fever [8]. In Europe, the species was incriminated as the vector in outbreak of, among others, chikungunya in 2007 northeastern Italy [9], dengue in 2010 in Croatia [10], dengue and Zika cases between 2010 and 2020 in France [11–13].

The first report of the tiger mosquito in the European continent dates back to 1979 in Albania [14]. It is thought to have been imported in shipments and containers from China in the mid-1970s. Later it has been detected in Italy in 1990 and has spread throughout the European continent via various routes [1]. In the surroundings of Kosovo, this species was recorded in Montenegro [15], in North Macedonia [16], and Serbia [17]. In Serbia, *Ae. albopictus* was intercepted in two districts in the western and southwestern part of the country. It has been present for the past nine years on the Croatian border (Batrovci, northwest of Serbia) [18], and on the Montenegro border since 2014 [19].

Distribution models predict that *Ae. albopictus* will continue to expand, depending on transport, environmental, and climatic changes [20–22].

This worldwide expansion is mostly based on transport and dissemination of dormant egg via the international trade of used tires [23] and shipments of the Asian plant "lucky bamboo" (*Dracaena spp.*) [24, 25] and by public and private ground transport from heavily infested areas [22]. In 2017, a research for tiger mosquito in Kosovo was conducted at the borders with Macedonia and Albania within the VectorNet project framework. It resulted negative, although Kosovo showed favorable conditions for the development of this mosquito species [26].

The primary objective of our study was, following a citizen report, to confirm the presence of the tiger mosquito in the territory of Kosovo and to determine its distribution.

## Materials and methods

### Study area

The present study was conducted in the municipality of Prizren (42.2166 N, 20.7333 E) and in the city of Suhareka (42.2248 N, 20.2248 E) in July, August and September 2020. The Municipality of Prizren, occupies the southern position in the Dukagjini Plain and southwestern Kosovo. The average altitude is about 450 m above sea level and includes nearly 640 km$^2$ of the Kosovo's entire surface. Sharri mountain, Prizren plain and downstream area of the Drini I Bardhë define the relief. Sharri mountain is also an essential geographical element since it constitutes a watershed between the Adriatic Sea and the Aegean Sea basin. The climate of this part of Kosovo is classified as Mediterranean. The city of Suhareka is located in the southern part of Kosovo. The average altitude is about 455 m. It is characterized by a rich hydrography.

### Collection and identification

We used ovitraps which are commonly used for detecting the females' presence via egg laying [27], and entomological aspirator (GeniccoSrl, Italy, Model: JF0825S1H—R) and BG-Sentinel™ traps (Biogents, Germany) to collect the adult mosquitoes. The ovitraps (500 ml black plastic cups) were filled with tap water and equipped with a masonite strip (12.5 × 2.5 cm) for egg deposition. The plastic cups were modified by two holes, punching 3 cm from the top of the cup to prevent water overfilling. A total of 36 ovitraps were randomly distributed in 7 localities [Vërmicë (5), Zhur (11), Vlashnje (5), Prizren (8), Atmaxhë (3), Landovicë (3) and Suharekë (1)]. The distance between the traps was 100 m at minimum. At four localities (Vërmicë, Zhur st.1, Zhur st.2, Prizren) we also used BG-Sentinel traps baited with BG-Lure and $CO_2$. The ovitraps were placed on the ground, in shaded and accessible places, under vegetation, with free space above at least 1 m (Fig 1A–1D).

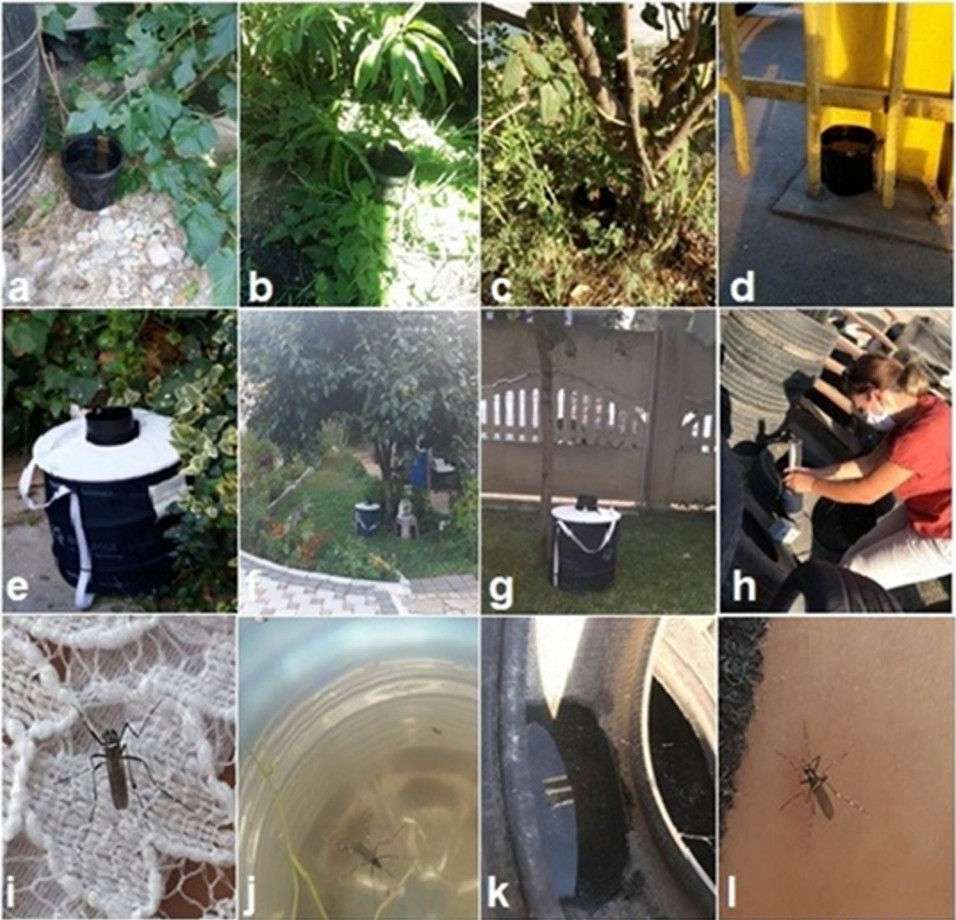

**Fig 1. Examples of traps operating at several mosquito collection sites.** Ovitraps in: a Vlashnje (Tyrecentres); b Zhur st.1 (Privat residence garden); c Prizren (Privat residence garden); d Vërmicë (Restaurant veranda); e Odour-baited adult traps (BG-Sentinel) Vermice (Restaurant garden); f Zhur st.1 (Resident garden); g Prizren (Resident garden). Catching with aspirator: h Vlashnje (Tyrecentres). Adults resting in: i Prizren (on the human body); j Zhur (Plastic bottle); k Vlashnje (Inside the tire, resting on the surface water); l Zhur st.1 (first specimen of *Aedes albopictus* caught while landing on human body).

The ovitraps were left at the same place during a sampling period of 10 successive days. After 10 days, the masonite strips were collected from the traps and transferred to the laboratory at the university of Prishtina for identification based on their color, size, shape and surface sculpting [28]. The adult mosquitoes were identified using the MosKeyTool identification key [29]. Another technique used in this research was aspiration from human bait. This technique was used in three localities (Zhur st.1, Vlashnje, Prizren, (Table 1) for 30 minutes (Fig 1H, 1I and 1L).

One BG-Sentinel trap was run for two consecutive nights every two weeks and for three periods (Table 1). The traps were set in the private houses' backyard (Fig 1E–1G). In total we did perform 12 BG-Sentinel trap-nights.

## Results

The presence of *Ae. albopictus* was registered in Kosovo for the first time at the end of July 2020 in Zhur village. The annoyance and bites that the mosquito caused to citizens during this period encouraged citizens to post a mosquito photo on a social media. This information

**Table 1. Results of *Aedes albopictus* adult trapping with BG-Sentinel traps (BG) or handheld electric aspirators (A) in the Prizreni municipality (July -September 2020).**

| Locality | Latitude (N) | Longitude (E) | Method of capture | Sampling area | Sampling period# | No. of adults | No. of adults |
|---|---|---|---|---|---|---|---|
| | | | | | | *Ae. albopictus* | *Cx.pipiens s.l* |
| Vërmicë | 42.166918 | 20.572473 | BG | Restaurant garden with vegetation | 25-07-20 | 0 | 7 |
| | | | | | 09-08-20 | 15 | 3 |
| | | | | | 25-08-20 | 12 | 3 |
| Zhur st.1 | 42.16604 | 20.61539 | BG | Private residence garden | 25-07-20 | 27 | 3 |
| | | | | | 09-08-20 | 10 | 0 |
| | | | | | 25-08-20 | 5 | 0 |
| Zhur st.2 | 42.161245 | 20.623351 | BG | Private residence garden | 25-07-20 | 0 | 2 |
| | | | | | 09-08-20 | 4 | 5 |
| | | | | | 25-08-20 | 0 | 0 |
| Prizren | 42.223997 | 20.734394 | BG | Private residence garden | 25-07-20 | 0 | 2 |
| | | | | | 09-08-20 | 2 | 5 |
| | | | | | 25-08-20 | 3 | 7 |
| Zhur st.1 | 42.168164 | 20.615606 | A | House yard | 01-08-20 | 1 | 0 |
| | | | | | 10-08-20 | 3 | 0 |
| Vlashnje | 42.198731 | 20.667758 | A | Tire storage | 03-09-20 | 9 | 0 |
| Prizren | 42.223997 | 20.734394 | A | House with vegetable garden | 30-08-20 | 1 | 0 |
| | | | | | 08-09-20 | 1 | 0 |

#Entomological survey period: I: 25–27 July 2020; II: 08–10 August 2020; III: 25–27 August 2020.

suggested the tiger mosquito to be present in this area and to perform a field survey. Our first catch, a male mosquito (Fig 2), was caught by hand on 23.07.2020 in a private residence garden at Zhur st.1 (Table 1). It was identified in the laboratory of the Institute of Public Health, Tirana, Albania, as belonging to *Ae. albopictus*.

In total, 52 ovitrap samples revealed positive, yielding 2,711 eggs collected at 36 sampling stations from 7 localities (Fig 3).

Using ovitraps, we collected eggs of *Ae. albopictus* but also of the native species *Aedes (Dahliana) geniculatus* (Olivier, 1791) (Table 2) at the Prizreni municipality. Eggs of *Ae. albopictus* were found at 22 out of the 36 sampling stations, while 14 remained negative. We did collect 440 eggs of *Ae. albopictus* at Vërmicë, 1,187 at Zhur, 139 at Vlashnje and 119 at Atmaxha, respectively. In the city of Prizren we collected 786 eggs from three stations, while 40 eggs were counted from ovitraps in 3 stations at a Truck Terminal, on the periphery of the city of Prizren. We didn't found any egg in the Landovica and Suhareka city (Table 2).

Eggs of *Ae. geniculatus* were found only at 3 sites from 3 locations, i.e. Vërmicë, Zhur and Prizren (Table 2).

Four adult females were collected with an aspirator at a house yard, nine adult mosquitoes at a tire center with numerous tires holding rainwater, (Fig 1h and 1K) and two males at a house vegetable garden. Using BG-Sentinel traps, we collected 78 adult mosquitoes (38 females and 39 males) at four sampling stations in gardens of residential houses or restaurant within three entomological survey periods (Table 1).

Forty-eight eggs were successfully hatched and were reared to adults (18 males / 30 females). All adult mosquitoes were morphologically identified and classified as *Ae. albopictus*. During the research period, we also identified 115 eggs of *Ae. geniculatus* caught with ovitraps (Table 2) and 34 adult mosquitoes of *Culex pipiens s.l.* Linnaeus, 1758 caught with BG-Sentinel trap. (Table 1).

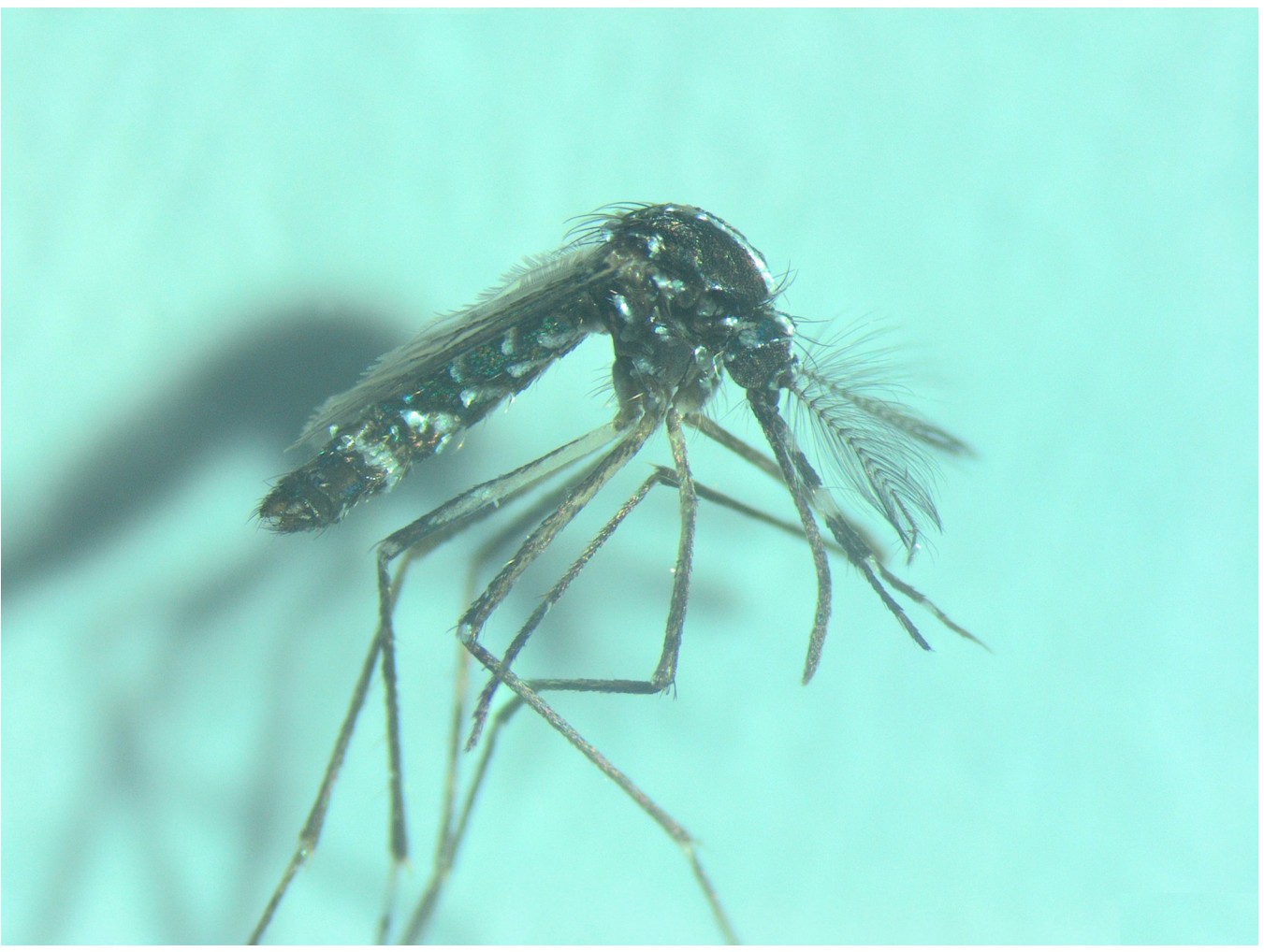

**Fig 2. First specimen of *Aedes albopictus* caught in Zhur st.1, date 23.07.2020 (photo: N. Muja-Bajraktari).**

## Discussion

The Asian tiger mosquito *Ae. albopictus*, originating from Southeast Asia, has undergone a significant expansion of its range in the last few decades [30].

In Europe, the tiger mosquito was first reported in Albania in 1979 [14], than ten years later in Italy [31], France [32, 33], Spain [34], Belgium [35], Switzerland [36], Greece [37], Montenegro [17, 38], Croatia [15], Bosnia and Herzegovina [15], Slovenia [15, 39] and North Macedonia [16]. The movement of cars has helped a lot in distributing *Ae. albopictus* species [40]. The first identification of the tiger mosquito was made near the border with Albania. It is thought that the way of its introduction was done through land routes strictly through the movement of vehicles.

Our study reports an established population of the Asian tiger mosquito in the municipality of Prizren in the southern part of Kosovo.

The ovitraps were selected as a research method due to their high sensitivity to low mosquito density, low price and practical use in the field [41]. A female tiger mosquito can lay eggs in several ovitraps placed in different areas; however, it depends on the sites' attractiveness [42]. Ovitraps can help control the mosquito population by eliminating the eggs, which results

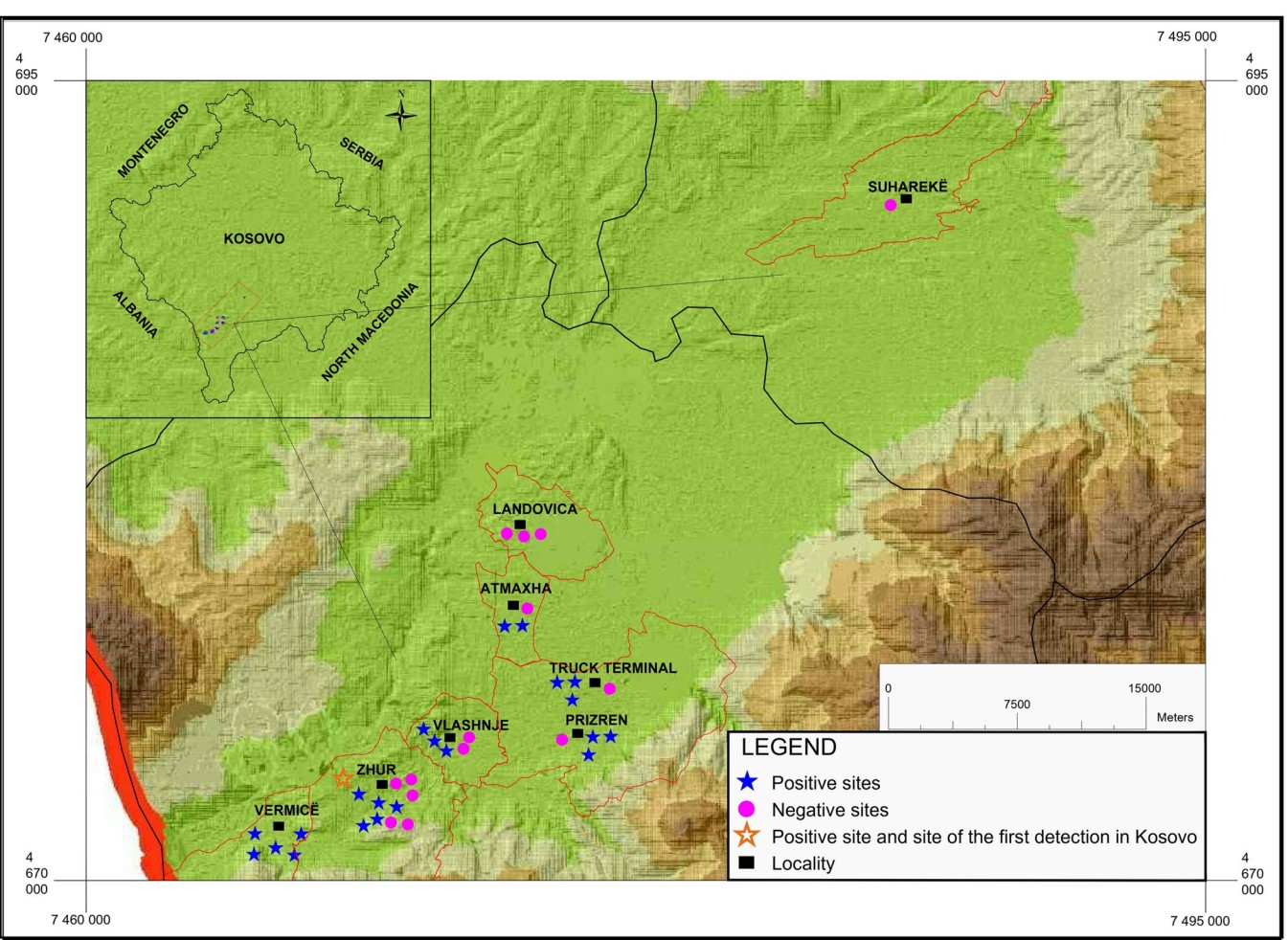

**Fig 3. Distribution map of the ovitraps in Prizreni region showing positive and negative site for the presence of *Aedes albopictus*.** The general administrative maps were extracted from the Natural Earth (https://www.naturalearthdata.com/) and then modified subsequently according to the information presented in the map, using Q-GIS version 3.18.

in a lower number of mosquitoes [16]. The first *Ae. albopictus* specimen is registered in Zhur, a village close to the border with Albania. This specimen recorded in a garden with many artificial breeding places (for container-breeding species).. The ability of the tiger mosquito to use both natural and artificial containers for larval habitats facilitates the widespread occupation of urban and peri urban environments [30], ensuring a close connection between the species and the human population and increasing the risk of vector-borne disease pathogen transmission in these areas [43].

The other sampling station, Vërmicë, is the closest to Albania's border, which argues the significant presence of eggs in the ovitraps and the large number of adults caught with BG-Sentinel traps. The border zones are considered highly vulnerable for introduction of potential invasive species. The number of cars and other vehicles coming from Albania stop for a considerable time at the border crossings, respectively in Vërmicë, so introducing the tiger mosquito through this road is indisputable. After that, from the border with Albania continues the highway, Morinë-Prizren-Prishtinë, therefore it is expected that mosquitoes will spread in other parts of Kosovo, precisely through the land route. Also, the third sampling station in Vlashnje is characterized by the significant presence of used tires, which served as breeding

**Table 2. Location of ovitraps in the Prizreni municipality and numbers of eggs collected for *Aedes albopictus* and *Aedes geniculatus*.**

| Localities | ID | Sampling period# | Latitude (N) | Longitude (E) | Sampling area | No. of eggs | |
|---|---|---|---|---|---|---|---|
| | | | | | | *Ae. albopictus* | *Ae. geniculatus* |
| Vërmicë | 01/1 | I, II, III | 42.167278 | 20.577672 | Near the road | 68 | 63 |
| | 01/2 | I, II, III | 42.168801 | 20.582681 | Near the road | 54 | 0 |
| | 01/3 | I, II, III | 42.166918 | 20.572473 | Restaurant garden | 202 | 0 |
| | 01/4 | I, II, III | 42.164574 | 20.567957 | Restaurant | 112 | 0 |
| | 01/5 | I, II, III | 42.163792 | 20.563543 | House veranda | 4 | 0 |
| Total | | | | | | 440 | 63 |
| Zhur | 02/1 | I, II, III | 42.168164 | 20.615606 | Private residence garden | 346 | 0 |
| | 02/2 | I, II, III | 42.166543 | 20.607746 | Private residence garden with pets | 196 | 0 |
| | 02/3 | I, II, III | 42.161062 | 20.622807 | Private residence garden | 168 | 48 |
| | 02/4 | I, II, III | 42.15736 | 20.618477 | Private residence garden | 62 | 0 |
| | 03/1 | I, II, III | 42.165116 | 20.623798 | Private residence garden | 175 | 0 |
| | 03/2 | I, II, III | 42.161867 | 20.611347 | Private residence garden | 240 | 0 |
| | 03/3 | I, II, III | 42.223801 | 20.734074 | Private residence garden | 0 | 0 |
| | 03/4 | I, II, III | 42.15956 | 20.629809 | Private residence garden | 0 | 0 |
| | 04/1 | IV | 42.163807 | 20.632446 | Private residence garden with chicken | 0 | 0 |
| | 04/2 | IV | 42.166677 | 20.627916 | Private residence garden | 0 | 0 |
| | 04/3 | IV | 42.15726 | 20.624888 | Near forest | 0 | 0 |
| Total | | | | | | 1,187 | 48 |
| Vlashnje | 05/1 | II, III | 42.200968 | 20.660929 | At the gas station near the road | 76 | 0 |
| | 05/2 | II, III | 42.204696 | 20.659614 | Inside the tires | 30 | 0 |
| | 05/3 | II, III | 42.201094 | 20.664804 | Inside the tires | 33 | 0 |
| | 05/4 | II, III | 42.203438 | 20.668572 | Tire center garden | 0 | 0 |
| | 05/5 | II, III | 42.200968 | 20.660929 | Tire center garden | 0 | 0 |
| Total | | | | | | 139 | 0 |
| Prizren | 06/1 | I, II, III, IV | 42.223288 | 20.741278 | Private residence garden | 438 | 4 |
| | 06/2 | I, II, III | 42.228297 | 20.739493 | Private residence garden | 176 | 0 |
| | 06/3 | I, II, III | 42.226967 | 20.729043 | Private residence garden | 172 | 0 |
| | 06/4 | I, II, III | 42.223801 | 20.734074 | Private residence garden | 0 | 0 |
| Total | | | | | | 786 | 4 |
| Truck terminal | 07/1 | IV | 42.249954 | 20.729769 | Truck terminal garden | 3 | 0 |
| | 07/2 | IV | 42.256972 | 20.733839 | Restaurant garden | 16 | 0 |
| | 07/3 | IV | 42.254285 | 20.72666 | Near the road | 0 | 0 |
| | 07/4 | IV | 42.250268 | 20.721548 | Truck terminal garden | 21 | 0 |
| Total | | | | | | 40 | 0 |
| Atmaxhë Total | 08/1 | III | 42.245329 | 20.699713 | At the gas station near the road | 73 | 0 |
| | 08/2 | III | 42.24801 | 20.696709 | Near the car wash | 46 | 0 |
| | 08/3 | III | 42.243715 | 20.702604 | Hotel garden | 0 | 0 |
| | | | | | | 119 | 0 |
| Landovicë | 09/1 | IV | 42.259853 | 20.688001 | Supermarket forecourt | 0 | 0 |
| | 09/2 | IV | 42.264082 | 20.6842 | Near the road | 0 | 0 |
| | 09/3 | IV | 42.255862 | 20.684525 | Private residence garden | 0 | 0 |
| Suharekë | 10/1 | I, II, III, IV | 42.363973 | 20.832803 | Bus station | 0 | 0 |
| Overall total | | | | | | 2,711 | 115 |

#Entomological survey period: I: 23–02 July 2020; II: 02–12 August 2020; III: 12–22 August 2020; IV: 02–12 September 2020.

sites. At the monitoring station inside the city of Prizren, the number of eggs in ovitraps was high, suggesting presence of a high number of adults. The recording of *Ae. albopictus* in this part of Kosovo is an important finding, demarcating new boundaries of the distribution range of the species in the Balkans and suggesting an increasing mosquito-borne disease threat to public health in that region. Based on our results, *Ae. albopictus* may be, by 2020, absent from the Landovica locality, in the periphery of Prizren, and from Suhareka, located about 41 km away from the Albanian border.

As a potential invasive species, it is expected that *Ae. albopictus* will expand its distribution to other parts of the country. This already happened with other invasive insect species recorded in Kosovo like the western conifer seed bug *Leptoglossus occidentalis*. This insect was reported from two localities in the east of the country, in the area of Batllava Lake, as well as in two localities from the foothills of Bjeshkët e Nemuna Mountains, in the western part of Kosovo [44].

## Conclusions

Since the presence of the tiger mosquito in the Republic of Kosovo is confirmed, it is expected that the Health authorities will develop a monitoring system for both mosquito and pathogen surveillance to survey and prevent the risk of pathogen transmission. Community involvement has been very successful in other countries, therefore we suggest to further encourage community involvement (citizen scientists) for an early detection of this invasive mosquito species in other parts of the country.

## Supporting information

**S1 Table. Results of the investigation for the presence of *Ae. albopictus* eggs at 36 stations in Kosovo and *Ae. albopictus* adult trapping with BG Sentinel traps (BG) or handheld electric aspirator (A).**
(XLSX)

## Acknowledgments

The work was done within the framework of AIM-COST Action CA17108.

We would like to express our gratitude to Prof. Dr Silvia Bino, Head of the Department of Epidemiology and Control of Infectious Diseases, Institute of Public Health in Tirana, Albania, for her continuous support of our research and Mihaela Kavran that provided papers for research conducted on Serbia.

We are grateful to Mr. Kreshnik Morina from Prishtina, Kosovo, who assisted us to prepare the new map of the Fig 3 of this manuscript. We also want to thank in particular Ms. Hatixhe Çollaku from Zhur who posted in social media the photo of the first specimen of *Aedes albopictus* and Mr. Artan Muja and all citizens for their cooperation to provide information on mosquito's presence and for giving us permission to investigate in their properties during the fieldwork.

## Author Contributions

**Conceptualization:** Nesade Muja-Bajraktari, Përparim Kadriaj, Ferdije Zhushi-Etemi, Kurtesh Sherifi, Bulent Alten, Dusan Petrić, Enkelejda Velo, Francis Schaffner.

**Data curation:** Nesade Muja-Bajraktari, Ferdije Zhushi-Etemi, Kurtesh Sherifi, Dusan Petrić, Enkelejda Velo.

**Formal analysis:** Nesade Muja-Bajraktari, Përparim Kadriaj, Ferdije Zhushi-Etemi, Kurtesh Sherifi, Bulent Alten, Dusan Petrić, Enkelejda Velo, Francis Schaffner.

**Investigation:** Nesade Muja-Bajraktari, Përparim Kadriaj, Kurtesh Sherifi, Bulent Alten, Dusan Petrić, Enkelejda Velo, Francis Schaffner.

**Methodology:** Nesade Muja-Bajraktari, Përparim Kadriaj, Ferdije Zhushi-Etemi, Kurtesh Sherifi, Bulent Alten, Dusan Petrić, Enkelejda Velo, Francis Schaffner.

**Resources:** Nesade Muja-Bajraktari, Përparim Kadriaj, Bulent Alten, Dusan Petrić, Enkelejda Velo, Francis Schaffner.

**Supervision:** Ferdije Zhushi-Etemi, Francis Schaffner.

**Validation:** Nesade Muja-Bajraktari, Përparim Kadriaj, Ferdije Zhushi-Etemi, Kurtesh Sherifi, Bulent Alten, Dusan Petrić, Enkelejda Velo, Francis Schaffner.

**Visualization:** Nesade Muja-Bajraktari, Përparim Kadriaj, Kurtesh Sherifi, Bulent Alten, Dusan Petrić, Enkelejda Velo, Francis Schaffner.

**Writing – original draft:** Nesade Muja-Bajraktari, Përparim Kadriaj.

**Writing – review & editing:** Ferdije Zhushi-Etemi, Kurtesh Sherifi, Bulent Alten, Dusan Petrić, Enkelejda Velo, Francis Schaffner.

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
