## [Decision Letter · Decision Letter 0]

29 Nov 2021

PONE-D-21-33477The Asian tiger mosquito Aedes albopictus (Skuse) in Kosovo: first recordPLOS ONE

Dear Dr. Velo,

Thank you for submitting your manuscript to PLOS ONE. After careful consideration, we feel that it has merit but does not fully meet PLOS ONE’s publication criteria as it currently stands. Therefore, we invite you to submit a revised version of the manuscript that addresses the points raised during the review process.

When preparing your revised manuscript, you are asked to carefully consider the reviewer comments which can be found below. In addition, you will need to check and re-format the reference list to make sure it conforms to the prerequisites of PLOS ONE. **The English language needs further improvement before publication**.

We look forward to receiving your revised manuscript.

Kind regards,

Carina Zittra, Ph.D., Mag.rer.nat.

Academic Editor

PLOS ONE

Journal Requirements:

3. We note that Figure 3 in your submission contain [map/satellite] images which may be copyrighted. All PLOS content is published under the Creative Commons Attribution License (CC BY 4.0), which means that the manuscript, images, and Supporting Information files will be freely available online, and any third party is permitted to access, download, copy, distribute, and use these materials in any way, even commercially, with proper attribution. For these reasons, we cannot publish previously copyrighted maps or satellite images created using proprietary data, such as Google software (Google Maps, Street View, and Earth). For more information, see our copyright guidelines: http://journals.plos.org/plosone/s/licenses-and-copyright.

a. You may seek permission from the original copyright holder of Figure 3 to publish the content specifically under the CC BY 4.0 license.  

4. We note that Figure 1 includes an image of a [patient / participant / in the study]. 

Reviewers' comments:

Reviewer's Responses to Questions

**Comments to the Author**

1. Is the manuscript technically sound, and do the data support the conclusions?

Reviewer #1: Yes

Reviewer #2: Yes

2. Has the statistical analysis been performed appropriately and rigorously? 

Reviewer #1: I Don't Know

Reviewer #2: N/A

3. Have the authors made all data underlying the findings in their manuscript fully available?

Reviewer #1: Yes

Reviewer #2: Yes

4. Is the manuscript presented in an intelligible fashion and written in standard English?

Reviewer #1: No

Reviewer #2: Yes

5. Review Comments to the Author

Reviewer #1: The authors describe the first records of tiger mosquitoes in Kosovo. The manuscript is of high relevance in the field and provides novel information.

Major comment: English language editing is needed.

Minor comments:

entire document – e.g. Line 31: Change Aedes to Ae. when needed (not at first mentioning and at beginning of sentences)

The definition “invasive” should be used with care. It has not proven to outcompete native mosquitoes in Kosovo. It is recommended to change to “potential invasive”

Line 64: Mosquitoes transmit pathogens and not diseases.

Line 82: Dracaena in italics

Line 109: The link to BG is not necessary. It is better to mention BG-sentinel generation 2. Moreover it is not mentioned if CO2 and lure were used. Info is give in line 139 – recommended to move to line 109

Line 182: s.l. not in italics

Line 180-190: Many countries are missing – if all European countries where Ae. albopictus was reported are planned to be listed

Mosquito alert is mentioned but no data shown. Maybe discuss that this app might increase the number of reports (the reviewer has seen tiger mosquitoes from the Kosovo when specifying for Mosquito Alert)

Reviewer #2: The manuscript is well written and provides interesting findings from Kosovo. There are some issues however to be addressed and most of them are commented directly in the manuscript pdf document. The abstract is somehow too long, and it would be nice to be shortened. In Introduction or somewhere in the Discussion section it would be advisable to give an overview of other invasive insects known from Kosovo, so that the paper is put also in the context of broader invasion challenge. English must be improved as well, although most of the linguistic corrections are done inside the document. I suggest to publish the paper after these revisions.

6. PLOS authors have the option to publish the peer review history of their article (what does this mean?). If published, this will include your full peer review and any attached files.

Reviewer #1: No

Reviewer #2: **Yes: **Halil Ibrahimi

---

## [Author Response · Author response to Decision Letter 0]

5 Feb 2022

Response to Reviewers

Dear reviewers, 

Thank you very much for the given comments. We find them very useful and tried to answer to each of them. 

Answer to Journal requirements

https://journals.plos.org/plosone/s/file?id=wjVg/PLOSOne_formatting_sample_main_body.pdfand

Answer: Manuscript is corrected according to the Journal requirements

Answer: Since the village Zhur where the first specimen of Aedes albopictus was recorded does not belong to a protected area, there was no need to ask for the permit to conduct our research. There was no need for permit to carry out the study also for the other localities. Citizens were very collaborative and allowed us to collect in their properties. 

3. We note that Figure 3 in your submission contain [map/satellite] images which may be copyrighted. All PLOS content is published under the Creative Commons Attribution License (CC BY 4.0), which means that the manuscript, images, and Supporting Information files will be freely available online, and any third party is permitted to access, download, copy, distribute, and use these materials in any way, even commercially, with proper attribution. For these reasons, we cannot publish previously copyrighted maps or satellite images created using proprietary data, such as Google software (Google Maps, Street View, and Earth). For more information, see our copyright guidelines: http://journals.plos.org/plosone/s/licenses-and-copyright.

Answer: A new map has been designed by another author for the purpose to be used in our Manuscript according to the PLOS ONE policy. 

The author of the new map, Mr. Kreshnik Morina (Geodesic) submitted a confirmation about the authorship and granted the permission to the authors of the Manuscript to use the content of the map for publication. He also explained that the general administrative maps were extracted from the Natural Earth (a public domain map dataset) https://www.naturalearthdata.com/ which is free to use in any type of project and then modified subsequently according to the information presented in the map, using Q-GIS version 3.18 (an open source GIS program).

4. We note that Figure 1 includes an image of a [patient / participant / in the study]. 

Answer: In the image of the figure is the first author of the manuscript during the research in the field. She included her photo in the Manuscript.

Answer: The references are checked carefully.

Reviewers' comments:

Reviewer's Responses to Questions

Comments to the Author

1. Is the manuscript technically sound, and do the data support the conclusions?

Reviewer #1: Yes

Reviewer #2: Yes

2. Has the statistical analysis been performed appropriately and rigorously?

Reviewer #1: I Don't Know

Reviewer #2: N/A

3. Have the authors made all data underlying the findings in their manuscript fully available?

Reviewer #1: Yes

Reviewer #2: Yes

4. Is the manuscript presented in an intelligible fashion and written in standard English?

Reviewer #1: No

Reviewer #2: Yes

5. Review Comments to the Author

Reviewer #1: The authors describe the first records of tiger mosquitoes in Kosovo. The manuscript is of high relevance in the field and provides novel information.

Major comment: English language editing is needed.

Answer: The English editing is done as suggested.

Minor comments:

entire document – e.g. Line 31: Change Aedes to Ae. when needed (not at first mentioning and at beginning of sentences)

Answer: It is changed as suggested

The definition “invasive” should be used with care. It has not proven to outcompete native mosquitoes in Kosovo. It is recommended to change to “potential invasive”

Answer: It is corrected. In the revised text is used potential invasive.

Line 64: Mosquitoes transmit pathogens and not diseases.

Answer: In the line 61 it is stated that Ae. albopictushas potential to transmit 22 arboviruses-pathogens, of the family Flaviviridae

Line 82: Dracaena in italics

Answer: it is corrected

Line 109: The link to BG is not necessary. It is better to mention BG-sentinel generation 2. Moreover it is not mentioned if CO2 and lure were used. Info is give in line 139 – recommended to move to line 109

Answer: It is corrected as suggested by reviewer.

Line 182: s.l. not in italics.

Answer: Corrected.

Line 180-190: Many countries are missing – if all European countries where Ae. albopictus was reported are planned to be listed

Answer: Other European countries where Ae.albopictus is reported are added.

Mosquito alert is mentioned but no data shown. Maybe discuss that this app might increase the number of reports (the reviewer has seen tiger mosquitoes from the Kosovo when specifying for Mosquito Alert).

Answer: The first author of the Manuscript used Mosquito Alert to report first record of Ae. albipictus in her country.

Reviewer #2: The manuscript is well written and provides interesting findings from Kosovo. There are some issues however to be addressed and most of them are commented directly in the manuscript pdf document. 

The abstract is somehow too long, and it would be nice to be shortened.

Answer: Abstract is shortened as suggested.

In Introduction or somewhere in the Discussion section it would be advisable to give an overview of other invasive insects known from Kosovo, so that the paper is put also in the context of broader invasion challenge. 

Answer: As suggested by the reviewer, an overview of other insect invasive species reported from Kosovo is given in the Discussion.

English must be improved as well, although most of the linguistic corrections are done inside the document. I suggest publishing the paper after these revisions.

Answer: The language corrections are done. 

Beside the correction based on the reviewers’ comments and suggestions, some other minor correction are done in the Manuscript aiming to improve its quality.

6. PLOS authors have the option to publish the peer review history of their article (what does this mean?). If published, this will include your full peer review and any attached files.

Do you want your identity to be public for this peer review? For information about this choice, including consent withdrawal, please see our Privacy Policy.

Reviewer #1: No

Reviewer #2: Yes: HalilIbrahimi

---

## [Editor Report · Decision Letter 1]

9 Feb 2022

The Asian tiger mosquito Aedes albopictus (Skuse) in Kosovo: first record

PONE-D-21-33477R1

Dear Dr. Velo,

We’re pleased to inform you that your manuscript has been judged scientifically suitable for publication and will be formally accepted for publication once it meets all outstanding technical requirements.

Kind regards,

Carina Zittra, Ph.D., Mag.rer.nat.

Academic Editor

PLOS ONE
---

## [Editor Report · Acceptance letter]

1 Mar 2022

PONE-D-21-33477R1 

The Asian tiger mosquito *Aedes albopictus* (Skuse) in Kosovo: first record 

Dear Dr. Velo:

I'm pleased to inform you that your manuscript has been deemed suitable for publication in PLOS ONE. Congratulations! Your manuscript is now with our production department. 

Kind regards, 

on behalf of

Dr. Carina Zittra 

Academic Editor

PLOS ONE